# BIO-INSPIRED HASHING FOR UNSUPERVISED SIMILARITY SEARCH

## ABSTRACT

The fruit fly Drosophila's olfactory circuit has inspired a new locality sensitive hashing (LSH) algorithm, `FlyHash`. In contrast with classical LSH algorithms that produce low dimensional hash codes, `FlyHash` produces sparse high-dimensional hash codes and has also been shown to have superior empirical performance compared to classical LSH algorithms in similarity search. However, `FlyHash` uses random projections and cannot *learn* from data. Building on inspiration from `FlyHash` and the ubiquity of sparse expansive representations in neurobiology, our work proposes a novel hashing algorithm `BioHash` that produces sparse high dimensional hash codes in a *data-driven* manner. We show that `BioHash` outperforms previously published benchmarks for various hashing methods. Since our learning algorithm is based on a local and biologically plausible synaptic plasticity rule, our work provides evidence for the proposal that LSH might be a computational reason for the abundance of sparse expansive motifs in a variety of biological systems. We also propose a convolutional variant `BioConvHash` that further improves performance. From the perspective of computer science, `BioHash` and `BioConvHash` are *fast*, *scalable* and yield compressed binary representations that are useful for similarity search.

## 1 INTRODUCTION

Sparse expansive representations are ubiquitous in neurobiology. Expansion means that a high-dimensional input is mapped to an even higher dimensional secondary representation. Such expansion is often accompanied by a sparsification of the activations: dense input data is mapped into a sparse code, where only a small number of secondary neurons respond to a given stimulus.

A classical example of the sparse expansive motif is the Drosophila fruit fly olfactory system. In this case, approximately 50 projection neurons send their activities to about 2500 Kenyon cells (Turner et al., 2008), thus accomplishing an approximately 50x expansion. An input stimulus typically activates approximately 50% of projection neurons, and less than 10% Kenyon cells (Turner et al., 2008), providing an example of significant sparsification of the expanded codes. Another example is the rodent olfactory circuit. In this system, dense input from the olfactory bulb is projected into piriform cortex, which has 1000x more neurons than the number of glomeruli in the olfactory bulb. Only about 10% of those neurons respond to a given stimulus (Mombaerts et al., 1996). A similar motif is found in rat's cerebellum and hippocampus (Dasgupta et al., 2017).

From the computational perspective, expansion is helpful for increasing the number of classification decision boundaries by a simple perceptron (Cover, 1965) or increasing memory storage capacity in models of associative memory (Hopfield, 1982). Additionally, sparse expansive representations have been shown to reduce intrastimulus variability and the overlaps between representations induced by distinct stimuli (Sompolinsky, 2014). Sparseness has also been shown to increase the capacity of models of associative memory (Tsodyks & Feigelman, 1988).

The goal of the present work is to use this "biological" inspiration about sparse expansive motifs, as well as local Hebbian learning, for designing a novel hashing algorithm `BioHash` that can be used in similarity search. Below we describe the task, the algorithm, and demonstrate that `BioHash` improves retrieval performance on common benchmark datasets.

**Similarity search and LSH.** In similarity search, given a query $q \in \mathbb{R}^d$, a similarity measure $sim(q, x)$, and a database $X \in \mathbb{R}^{n \times d}$ containing $n$ items, the objective is to retrieve a ranked list of $R$ items from the database most similar to $q$. When data is high-dimensional (e.g. images/documents) and the databases are large (millions or billions items), this is a computationally challenging problem. However, approximate solutions are generally acceptable, with Locality Sensitive Hashing (LSH) being one such approach (Wang et al., 2014). Similarity search approaches maybe unsupervised or supervised. Since labelled information for extremely large datasets is infeasible to obtain, our work focuses on the unsupervised setting. In LSH (Indyk & Motwani, 1998; Charikar, 2002), the idea is to encode each database entry $x$ (and query $q$) with a binary representation $h(x)$ ($h(q)$ respectively) and to retrieve $R$ entries with smallest Hamming distances $d_H(h(x), h(q))$. Intuitively, (see (Charikar, 2002), for a formal definition), a hash function $h : \mathbb{R}^d \to \{-1, 1\}^m$ is said to be *locality sensitive*, if similar (dissimilar) items $x_1$ and $x_2$ are close by (far apart) in Hamming distance $d_H(h(x_1), h(x_2))$. LSH algorithms are of fundamental importance in computer science, with applications in similarity search, data compression and machine learning (Andoni & Indyk, 2008).

**Drosophila olfactory circuit and FlyHash.** In classical LSH approaches, the data dimensionality $d$ is much larger than the embedding space dimension $m$, resulting in low-dimensional hash codes (Wang et al., 2014; Indyk & Motwani, 1998; Charikar, 2002). In contrast, a new family of hashing algorithms has been proposed (Dasgupta et al., 2017) where $m \gg d$, but the secondary representation is highly sparse with only a small number $k$ of $m$ units being active, see Figure 1. We call this algorithm `FlyHash` in this paper, since it is motivated by the computation carried out by the fly's olfactory circuit. The expansion from the $d$ dimensional input space into an $m$ dimensional secondary representation is carried out using a random set of weights $W$ (Dasgupta et al., 2017; Caron et al., 2013). The resulting high dimensional representation is sparsified by $k$-Winner-Take-All ($k$-WTA) feedback inhibition in the hidden layer resulting in top $\sim 5\%$ of units staying active (Lin et al., 2014; Stevens, 2016).

While `FlyHash` uses random synaptic weights, sparse expansive representations are not necessarily random (Sompolinsky, 2014), perhaps not even in the case of Drosophila (Gruntman & Turner, 2013; Zheng et al., 2018). Moreover, using synaptic weights that are learned from data might help to further improve the locality sensitivity property of `FlyHash`. Thus, it is important to investigate the role of learned synapses on the hashing performance. A recent work `SOLHash` (Li et al., 2018), takes inspiration from `FlyHash` and attempts to adapt the synapses to data, demonstrating improved performance over `FlyHash`. However, every learning update step in `SOLHash` invokes a constrained linear program and also requires computing pairwise inner-products between all training points, making it very time consuming and limiting its scalability to datasets of even modest size. These limitations restrict `SOLHash` to training only on a small fraction of the data (Li et al., 2018). Additionally, `SOLHash` is biologically implausible (for an extended discussion, see Sec. 5). `BioHash` also takes inspiration from `FlyHash` and demonstrates improved performance compared to random weights used in `FlyHash`, but it is fast, online, scalable and, importantly, `BioHash` is neurologically plausible.

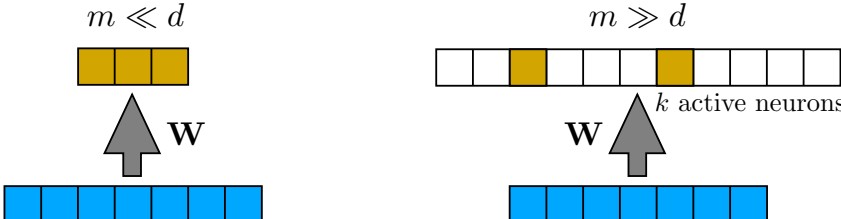

Figure 1: It is convenient to think about different hashing algorithms in terms of representational contraction (large dimension of input is mapped into a smaller dimensional latent space), or expansion (large dimensional input is mapped into an even larger dimensional latent space). The projections can be random or data driven.

Not only "biological" inspiration can lead to improving hashing techniques, but the opposite might also be true. One of the statements of the present paper is that `BioHash` satisfies locality sensitive property, and, at the same time, utilizes a biologically plausible learning rule for synaptic weights (Krotov & Hopfield, 2019). This provides evidence toward the proposal that the reason why sparse expansive representations are so common in biological organisms is because they perform locality

sensitive hashing. In other words, they cluster similar stimuli together and push distinct stimuli far apart. Thus, our work provides evidence toward the proposal that LSH might be a fundamental computational principle utilized by the sparse expansive circuits Fig. 1 (right). Importantly, learning of synapses must be neurobiologically plausible (the synaptic plasticity rule should be local).

**Contributions.** Building on inspiration from `FlyHash` and more broadly the ubiquity of sparse, expansive representations in neurobiology, our work proposes a novel hashing algorithm `BioHash`, that in contrast with previous work (Dasgupta et al., 2017; Li et al., 2018), produces sparse high dimensional hash codes in a *data-driven* manner and with learning of synapses in a *neurobiologically plausible way*. We provide an existence proof for the proposal that LSH maybe a fundamental computational principle in neural circuits (Dasgupta et al., 2017) in the context of learned synapses. We incorporated convolutional structure into `BioHash`, resulting in a hashing with improved performance compared to previously published benchmarks. From the perspective of computer science, we show that `BioHash` is simple, scalable to large datasets and demonstrates good performance for similarity search. Interestingly, `BioHash` outperforms a number of recent SOTA deep hashing methods trained via backpropogation.

## 2 Approximate Similarity Search via BioHashing

Formally, if we denote a data point as $x \in \mathbb{R}^d$, we seek a binary hash code $y \in \{-1, 1\}^m$. We define the hash length of a binary code as $k$, if the exact Hamming distance computation is $O(k)$. Below we present our bio-inspired hashing algorithm.

### 2.1 Bio-inspired Hashing (BioHash)

We adopt a biologically plausible unsupervised algorithm for representation learning from Krotov & Hopfield (2019). Denote the synapses from the input layer to the hash layer as $\mathbf{W} \in \mathbb{R}^{m \times d}$. The learning dynamics for the synapses of an individual neuron $\mu$, denoted by $W_{\mu i}$, is given by

$$\tau \frac{dW_{\mu i}}{dt} = g\Big[\text{Rank}\big(\langle W_\mu, x \rangle_\mu\big)\Big]\Big(x_i - \langle W_\mu, x \rangle_\mu W_{\mu i}\Big), \tag{1}$$

where $W_\mu = (W_{\mu 1}, W_{\mu 2}...W_{\mu d})$, and

$$g[\mu] = \begin{cases} 1, & \mu = 1 \\ -\Delta, & \mu = r \\ 0, & \text{otherwise} \end{cases} \tag{2}$$

and $\langle x, y \rangle_\mu = \sum_{i,j} \eta_{i,j}^\mu x_i y_j$, with $\eta_{i,j}^\mu = |W_{\mu i}|^{p-2}\delta_{ij}$, where $\delta_{ij}$ is Kronecker delta and $\tau$ is the time scale of the learning dynamics. The Rank operation in equation (1) sorts the inner products from the largest ($\mu = 1$) to the smallest ($\mu = m$). It can be shown that the synapses converge to a unit ($p-$norm) sphere (Krotov & Hopfield, 2019). The training dynamics can be shown to minimize the following energy function

$$E = -\sum_A \sum_{\mu=1}^m g\Big[\text{Rank}\big(\langle W_\mu, x^A \rangle_\mu\big)\Big] \frac{\langle W_\mu, x^A \rangle_\mu}{\langle W_\mu, W_\mu \rangle_\mu^{\frac{p-1}{p}}}, \tag{3}$$

where $A$ indexes the training example. Note that the training dynamics do not perform gradient descent, i.e $\dot{W}_\mu \neq \nabla_{W_\mu} E$. However, time derivative of the energy function under dynamics (1) is always negative (we show this for the case $\Delta = 0$ below),

$$\tau \frac{dE}{dt} = -\sum_A \frac{\tau(p-1)}{\langle W_{\hat\mu}, W_{\hat\mu} \rangle_{\hat\mu}^{\frac{p-1}{p}+1}} \Big[\langle \frac{dW_{\hat\mu}}{dt}, x^A \rangle_{\hat\mu} \langle W_{\hat\mu}, W_{\hat\mu} \rangle_{\hat\mu} - \langle W_{\hat\mu}, x^A \rangle_{\hat\mu} \langle \frac{dW_{\hat\mu}}{dt}, W_{\hat\mu} \rangle_{\hat\mu}\Big] =$$

$$= -\sum_A \frac{\tau(p-1)}{\langle W_{\hat\mu}, W_{\hat\mu} \rangle_{\hat\mu}^{\frac{p-1}{p}+1}} \Big[\langle x^A, x^A \rangle_{\hat\mu} \langle W_{\hat\mu}, W_{\hat\mu} \rangle_{\hat\mu} - \langle W_{\hat\mu}, x^A \rangle_{\hat\mu}^2\Big] \leq 0, \tag{4}$$

where Cauchy-Schwartz inequality is used. For every training example $A$ the index of the activated hidden unit is defined as

$$\hat\mu = \arg\max_\mu \big[\langle W_\mu, x^A \rangle_\mu\big]. \tag{5}$$

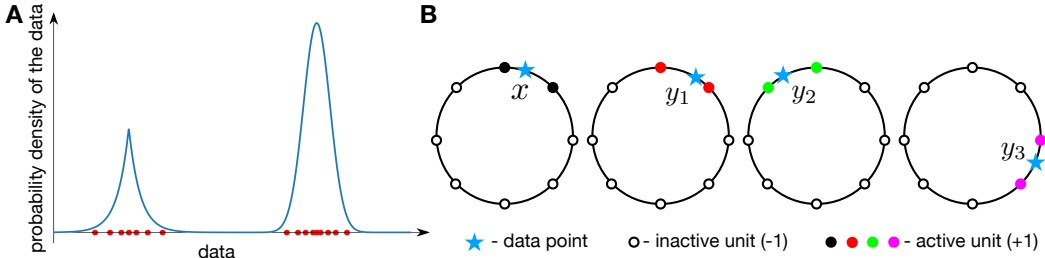

Figure 2: (Panel A) Distribution of the hidden units (red circles) for a given distribution of the data (in one dimension). (Panel B) Arrangement of hidden units for the case of homogeneous distribution of the training data $\rho = 1/(2\pi)$. For hash length $k = 2$ only two hidden units are activated (filled circles). If two data points are close to each other ($x$ and $y_1$) they elicit similar hash codes, if the two data points are far away from each other ($x$ and $y_3$) - the hash codes are different.

Thus, the energy function decreases during learning. A similar result can be shown for $\Delta \neq 0$.

After the learning-phase is complete, the hash code is generated, as in `FlyHash`, via WTA sparsification: for a given query $x$ we generate a hash code $y \in \{-1, 1\}^m$ as

$$y_\mu = \begin{cases} 1, & \langle W_\mu, x \rangle_\mu \text{ is in top } k \\ -1, & \text{otherwise.} \end{cases} \tag{6}$$

Thus, the hyperparameters of the method are $p, r, m$ and $\Delta$. Note that the synapses are updated based only on pre- and post-synaptic activations resulting in Hebbian or anti-Hebbian updates. Many "unsupervised" learning to hash approaches provide a sort of "weak supervision" in the form of similarities evaluated in the feature space of deep CNNs trained on ImageNet (Jin et al., 2019) to achieve good performance. `BioHash` does not assume such information is provided and is completely unsupervised.

## 2.2 INTUITION BEHIND THE LEARNING ALGORITHM

An intuitive way to think about the learning algorithm is to view the hidden units as particles that are attracted to local peaks of the density of the data, and that simultaneously repel each other. To demonstrate this, it is convenient to think about input data as randomly sampled points from a continuous distribution. Consider the case when $p = 2$ and $\Delta = 0$. In this case, the energy function can be written as (since for $p = 2$ the inner product does not depend on the weights, we drop the subscript $\mu$ of the inner product)

$$E = -\frac{1}{n} \sum_A \frac{\langle W_{\hat\mu}, x^A \rangle}{\langle W_{\hat\mu}, W_{\hat\mu} \rangle^{\frac{1}{2}}} = -\int \prod_i dv_i \frac{1}{n} \sum_A \delta(v_i - x_i^A) \frac{\langle W_{\hat\mu}, v \rangle}{\langle W_{\hat\mu}, W_{\hat\mu} \rangle^{\frac{1}{2}}} = \\ = -\int \prod_i dv_i \rho(v) \frac{\langle W_{\hat\mu}, v \rangle}{\langle W_{\hat\mu}, W_{\hat\mu} \rangle^{\frac{1}{2}}}, \tag{7}$$

where we introduced a continuous density of data $\rho(v)$. Furthermore, consider the case of $d = 2$, and imagine that the data lies on a unit circle. In this case the density of data can be parametrized by a single angle $\varphi$. Thus, the energy function can be written as

$$E = -\int_{-\pi}^{\pi} d\varphi \rho(\varphi) \cos(\varphi - \varphi_{\hat\mu}), \quad \text{where} \quad \hat\mu = \arg\max_\mu \left[ \cos(\varphi - \varphi_\mu) \right] \tag{8}$$

It is instructive to solve a simple case when the data follows an exponential distribution concentrated around zero angle with the decay length $\sigma$ ($\alpha$ is a normalization constant),

$$\rho(\varphi) = \alpha e^{-\frac{|\varphi|}{\sigma}}. \tag{9}$$

In this case, the energy (8) can be calculated exactly for any number of hidden units $m$. However, minimizing over the position of hidden units cannot be done analytically for general $m$. To further

simplify the problem consider the case when the number of hidden units $m$ is small. For $m = 2$ the energy is equal to

$$E = -\alpha \frac{\sigma(1 + e^{-\frac{\pi}{\sigma}})}{1 + \sigma^2} \Big( \cos(\varphi_1) + \sigma \sin(\varphi_1) + \cos(\varphi_2) - \sigma \sin(\varphi_2) \Big) \tag{10}$$

Thus, in this simple case the energy is minimized when

$$\varphi_{1,2} = \pm \arctan(\sigma) \tag{11}$$

In the limit when the density of data is concentrated around zero angle ($\sigma \to 0$) the hidden units are attracted to the origin and $|\varphi_{1,2}| \approx \sigma$. In the opposite limit ($\sigma \to \infty$) the data points are uniformly distributed on the circle. The resulting hidden units are then organized to be on the opposite sides of the circle $|\varphi_{1,2}| = \frac{\pi}{2}$, due to mutual repulsion.

Another limit when the problem can be solved analytically is the uniform density of the data $\rho = 1/(2\pi)$ for arbitrary number $m$ of hidden units. In this case the hidden units span the entire circle homogeneously - the angle between two consecutive hidden units is $\Delta\varphi = 2\pi/m$.

These results are summarized in an intuitive cartoon in Figure 2, panel A. After learning is complete, the hidden units, denoted by circles, are localized in the vicinity of local maxima of the probability density of the data. At the same time, repulsive force between the hidden units prevents them from collapsing onto the exact position of the local maximum. Thus the concentration of the hidden units near the local maxima becomes high, but, at the same time, they span the entire support (area where there is non-zero density) of the data distribution.

For hashing purposes, trying to find a data point $x$ "closest" to some new query $q$ requires a definition of "distance". Since this measure is wanted only for nearby locations $q$ and $x$, it need not be accurate for long distances. If we pick a set of $m$ reference points in the space, then the location of point $x$ can be specified by noting the few reference points it is closest to, producing a sparse and useful local representation. Uniformly tiling a high dimensional space is not a computationally useful approach. Reference points are needed only where there is data, and high resolution is needed only where there is high data density. The learning dynamics in 1 distributes $m$ reference vectors by an iterative procedure such that their density is high where the data density is high, and low where the data density is low. This is exactly what is needed for a good hash code.

The case of uniform density on a circle is illustrated in Figure 2, panel B. After learning is complete the hidden units homogeneously span the entire circle. For hash length $k = 2$, any given data point activates two closest hidden units. If two data points are located between two neighboring hidden units (like $x$ and $y_1$) they produce exactly identical hash codes with hamming distance zero between them (black and red active units). If two data points are slightly farther apart, like $x$ and $y_2$, they produce hash codes that are slightly different (black and green circles, hamming distance is equal to 2 in this case). If the two data points are even farther, like $x$ and $y_3$, their hash codes are not overlapping at all (black and magenta circles, hamming distance is equal to 4). Thus, intuitively similar data activate similar hidden units, resulting in similar representations, while dissimilar data result in very different hash codes.

## 2.3 COMPUTATIONAL COMPLEXITY AND METABOLIC COST.

In classical LSH algorithms (Charikar, 2002; Indyk & Motwani, 1998), typically, $k = m$ and $m \ll d$, entailing a storage cost of $k$ bits per database entry and $O(k)$ computational cost to compute Hamming distance. In `BioHash` (and in `FlyHash`), typically $m \gg k$ and $m > d$ entailing storage cost of $k \log_2 m$ bits per database entry and $O(k)$ [1] computational cost to compute Hamming distance. Note that while there is additional storage/lookup overhead over classical LSH in maintaining pointers, this is not unlike the storage/lookup overhead incurred by quantization methods like Product Quantization (PQ) (Jégou et al., 2011), which stores a lookup table of distances between every pair of codewords for each product space. From a neurobiological perspective, a highly sparse representation such as the one produced by `BioHash` keeps the same metabolic cost (Levy & Baxter, 1996) as a dense low-dimensional ($m \ll d$) representation, such as in classical `LSH` methods. At the same time, as we empirically show below, it better preserves similarity information.

---

[1]If we maintain sorted pointers to the locations of 1s, we have to compute the intersection between 2 ordered lists of length $k$, which is $O(k)$.

**Query**                    **Top Retrievals**

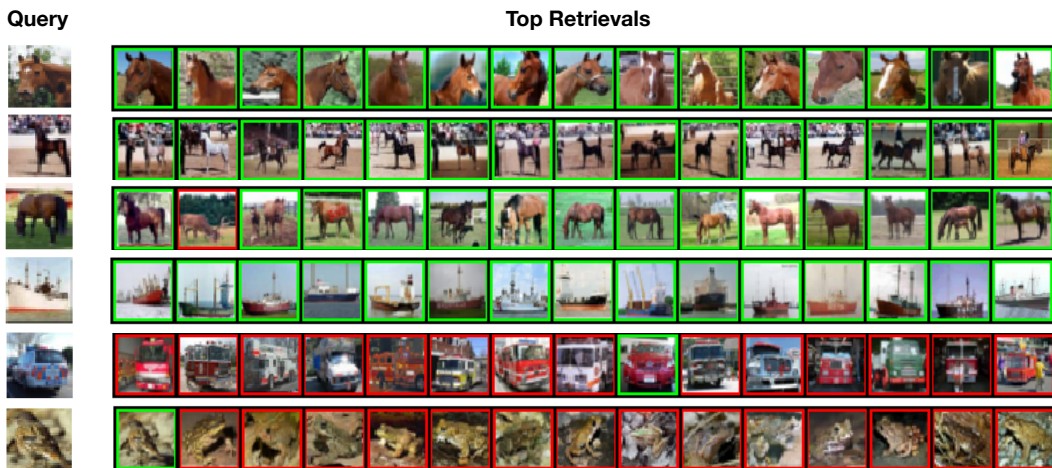

Figure 3: Examples of queries and top 15 retrievals using `BioHash` ($k = 16$) on VGG16 fc7 features of CIFAR-10. Retrievals have a green (red) border if the image is in the same (different) semantic class as the query image. We show some success (top 4) and failure (bottom 2) cases. However, it can be seen that even the failure cases are reasonable.

|  | Hash Length ($k$) | | | | | |
|---|---|---|---|---|---|---|
| Method | 2 | 4 | 8 | 16 | 32 | 64 |
| LSH | 12.45 | 13.77 | 18.07 | 20.30 | 26.20 | 32.30 |
| PCAHash | 19.59 | 23.02 | 29.62 | 26.88 | 24.35 | 21.04 |
| FlyHash | 18.94 | 20.02 | 24.24 | 26.29 | 32.30 | 38.41 |
| SH | 20.17 | 23.40 | 29.76 | 28.98 | 27.37 | 24.06 |
| ITQ | 21.94 | 28.45 | 38.44 | 41.23 | 43.55 | 44.92 |
| DH | - | - | - | 43.14 | 44.97 | 46.74 |
| UH-BNN | - | - | - | 45.38 | 47.21 | - |
| NaiveBioHash | 25.85 | 29.83 | 28.18 | 31.69 | 36.48 | 38.50 |
| BioHash | 44.38 | 49.32 | 53.42 | 54.92 | 55.48 | - |
| BioConvHash | **64.49** | **70.54** | **77.25** | **80.34** | **81.23** | - |

Table 1: mAP@All (%) on MNIST (higher is better). Best results (second best) for each hash length are in **bold** (underlined). `BioHash` demonstrates the best retrieval performance, substantially outperforming other methods including deep hashing methods `DH` and `UH-BNN`, especially at small $k$. Performance for `DH` and `UH-BNN` is unavailable for some $k$, since it is not reported in the literature.

## 2.4 CONVOLUTIONAL BIOHASH

In order to take advantage of the spatial statistical structure present in images, we use the dynamics in 1 to learn convolutional filters by training on image patches as in Grinberg et al. (2019). Convolutions in this case are unusual since the patches of the images are normalized to be unit vectors before calculating the inner product with the filters. Differently from Grinberg et al. (2019), we use *cross channel inhibition* to suppress the activities of the hidden units that are weakly activated. Specifically, if there are $F$ convolutional filters, then only the top $k_{CI}$ of $F$ activations are kept active per spatial location.

Patch normalization is reminiscent of the canonical computation of divisive normalization (Carandini & Heeger, 2011) and performs local intensity normalization. This is not unlike divisive normalization in the fruit fly's projection neurons. Divisive normalization has also been found to be beneficial (Ren et al., 2016) in CNNs trained end-to-end by the backpropagation algorithm on a supervised task. We found that the cross channel inhibition is important for a good hashing performance. Post cross-channel inhibition, we use a max-pooling layer, followed by a `BioHash` layer as in Sec. 2.1.

| Method | Hash Length ($k$) | | | | | |
|---|---|---|---|---|---|---|
| | 2 | 4 | 8 | 16 | 32 | 64 |
| LSH | 11.73 | 12.49 | 13.44 | 16.12 | 18.07 | 19.41 |
| PCAHash | 12.73 | 14.31 | 16.20 | 16.81 | 17.19 | 16.67 |
| FlyHash | 14.62 | 16.48 | 18.01 | 19.32 | 21.40 | 23.35 |
| SH | 12.77 | 14.29 | 16.12 | 16.79 | 16.88 | 16.44 |
| ITQ | 12.64 | 14.51 | 17.05 | 18.67 | 20.55 | 21.60 |
| NaiveBioHash | 11.79 | 12.43 | 14.54 | 16.62 | 17.75 | 18.65 |
| BioHash | 20.47 | 21.61 | 22.61 | 23.35 | 24.02 | - |
| BioConvHash | **26.94** | **27.82** | **29.34** | **29.74** | **30.10** | - |

Table 2: mAP@1000 (%) on CIFAR-10 (higher is better). Best results (second best) for each hash length are in **bold** (underlined). `BioHash` demonstrates the best retrieval performance, especially at small $k$.

## 3 SIMILARITY SEARCH

In this section, we empirically evaluate `BioHash`, investigate the role of sparsity in the latent space, and compare our results with previously published benchmarks. We consider two settings for evaluation: a) the training set contains unlabeled data, and the labels are only used for the evaluation of the performance of the hashing algorithm and b) where supervised pretraining on a different dataset is permissible. Features extracted from this pretraining are then used for hashing. In both settings `BioHash` outperforms previously published benchmarks for various hashing methods.

### 3.1 EVALUATION METRIC

Following previous work (Dasgupta et al., 2017; Li et al., 2018; Su et al., 2018), we use Mean Average Precision (mAP) as the evaluation metric, a measure that averages precision over different recall. Specifically, given a query set $Q$, we evaluate mAP@$R$ as

$$\text{mAP@}R \stackrel{\text{def}}{=} \frac{1}{|Q|} \sum_{q=1}^{|Q|} \frac{1}{\sum \text{Rel}(l)} \sum_{l=1}^{R} \text{Precision}(l)\text{Rel}(l), \tag{12}$$

where $R$ is number of retrievals, $\text{Rel}(l) = \mathbb{1}(\text{document } l \text{ is relevant})$ and $\text{Precision}(l)$ is the precision of the top $l$ retrievals. Notation: when $R$ is equal to size of the entire database, i.e a ranking of the entire database is desired, we use the notation mAP@All or simply mAP, dropping the reference to $R$.

### 3.2 DATASETS AND PROTOCOL

To make our work comparable with recent related work, we used common benchmark datasets: a) MNIST (Lecun et al., 1998), a dataset of 70k grey-scale images (size 28 x 28) of hand-written digits with 10 classes of digits ranging from "0" to "9", b) CIFAR-10 (Krizhevsky, 2009), a dataset containing 60k images (size 32x32x3) from 10 classes (e.g: car, bird).

Following the protocol in Lu et al. (2017); Chen et al. (2018), on MNIST we randomly sample 100 images from each class to form a query set of 1000 images. We use the rest of the 69k images as the training set for `BioHash` as well as the database for retrieval post training. Similarly, on CIFAR-10, following previous work (Su et al., 2018; Chen et al., 2018; Jin, 2018), we randomly sampled 1000 images per class to create a query set containing 10k images. The remaining 50k images were used for training as well as the database for retrieval as in the case of MNIST. Ground truth relevance for both dataset is based on class labels. Following previous work (Chen et al., 2018; Lin et al., 2015; Jin et al., 2019), we use mAP@1000 for CIFAR-10 and mAP@All for MNIST. It is common to benchmark the performance of hashing methods at hash lengths $k \in \{16, 32, 64\}$. However, it was observed in Dasgupta et al. (2017) that the regime in which `FlyHash` outperformed `LSH` was in the regime of low hash lengths $k \in \{2, 4, 8, 16, 32\}$. Accordingly, we evaluate performance for $k \in \{2, 4, 8, 16, 32, 64\}$.

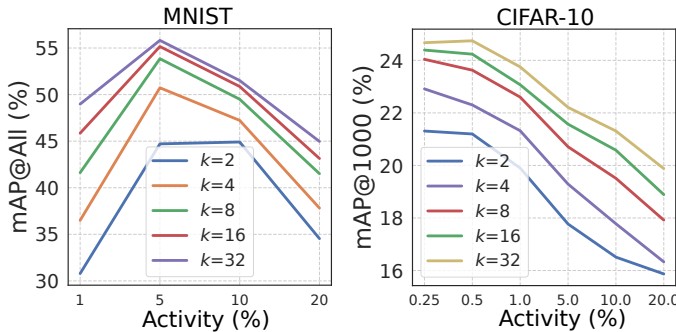

Figure 4: Effect of varying sparsity (activity): optimal activity % for MNIST and CIFAR-10 are 5% and 0.25%. Since the improvement in performance is small from 0.5 % to 0.25 %, we use 0.5% for CIFAR-10 experiments. The change of activity is accomplished by changing $m$ at fixed $k$.

### 3.3 BASELINES

As baselines we include random hashing methods `FlyHash` (Dasgupta et al., 2017), classical LSH (`LSH` (Charikar, 2002)), and data-driven hashing methods `PCAHash` (Gong & Lazebnik, 2011), Spectral Hashing (`SH` (Weiss et al., 2009)), Iterative Quantization (`ITQ` (Gong & Lazebnik, 2011)). As in Dasgupta et al. (2017), for `FlyHash` we set the sampling rating from PNs to KCs to be 0.1 and $m = 10d$. Additionally, where appropriate, we also compare performance of `BioHash` to deep hashing methods: `DeepBit` (Lin et al., 2015), `DH` (Lu et al., 2017), `USDH` (Jin et al., 2019), `UH-BNN` (Do et al., 2016), `SAH` (Do et al., 2017) and `GreedyHash` (Su et al., 2018). As previously discussed, in nearly all similarity search methods, a hash length of $k$ entails a dense representation using $k$ units. In order to clearly demonstrate the utility of sparse expansion in `BioHash`, we include a baseline (termed "`NaiveBioHash`"), which uses the learning dynamics in 1 but without sparse expansion, i.e the input data is projected into a dense latent representation with $k$ hidden units. The activations of those hidden units are then binarized to generate a hash code of length $k$.

### 3.4 RESULTS AND DISCUSSION

The performance of `BioHash` on MNIST is shown in Table 1. `BioHash` demonstrates the best retrieval performance, substantially outperforming other methods, including deep hashing methods `DH` and `UH-BNN`, especially at small $k$. Indeed, even at a very short hash length of $k = 2$, the performance of `BioHash` is comparable to or better than `DH` for $k \in \{16, 32\}$, while at $k = 4$, the performance of `BioHash` is better than the `DH` and `UH-BNN` for $k \in \{16, 32, 64\}$. The performance of `BioHash` saturates around $k = 16$, showing only a small improvement from $k = 8$ to $k = 16$ and an even smaller improvement from $k = 16$ to $k = 32$; accordingly, we do not evaluate performance at $k = 64$. We note that while `SOLHash` also evaluated retrieval performance on MNIST and is a data-driven hashing method inspired by Drosophila's olfactory circuit, the ground truth in their experiment was top 100 nearest neighbors of a query in the database, based on Euclidean distance between pairs of images in pixel space and thus cannot be directly compared[2]. Nevertheless, we adopt that protocol (Li et al., 2018) and show that `BioHash` substantially outperforms `SOLHash` in Sec 5.

The performance of `BioHash` on CIFAR-10 is shown in Table 2. Similar to the case of MNIST, `BioHash` demonstrates the best retrieval performance, substantially outperforming other methods, especially at small $k$. Even at $k \in \{2, 4\}$, the performance of `BioHash` is comparable to other methods with $k \in \{16, 32, 64\}$. This suggests that `BioHash` is a particularly good choice when short hash lengths are required.

**Effect of sparsity** For a given hash length $k$, we parametrize the total number of neurons $m$ in the hash layer as $m \times a = k$, where $a$ is the activity i.e the fraction of active neurons. For each hash length $k$, we varied % of active neurons and evaluated the performance on a validation set

---

[2]Due to missing values of the hyperparameters we are unable to reproduce the performance of `SOLHash` to enable a direct comparison.

| Activity (%): 1.0 | Activity (%): 5.0 | Activity (%): 20.0 | Activity (%): 70.0 |
|---|---|---|---|
| 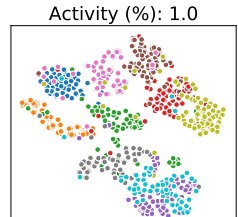 | 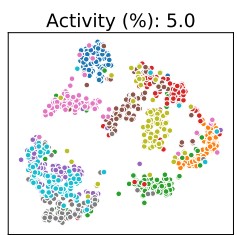 | 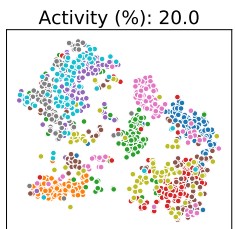 | 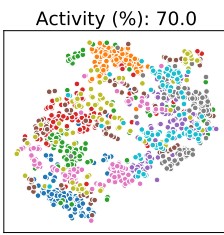 |

Figure 5: tSNE embedding of MNIST as the activity is varied for a fixed $m = 160$ (the change of activity is accomplished by changing $k$ at fixed $m$). When the sparsity of activations decreases (activity increases), some clusters merge together, though highly dissimilar clusters (e.g. orange and blue in the lower left) stay separated.

| | Hash Length ($k$) | | | | | | Hash Length ($k$) | | | |
|---|---|---|---|---|---|---|---|---|---|---|
| $k_{\text{CI}}$ | 2 | 4 | 8 | 16 | | $k_{\text{CI}}$ | 2 | 4 | 8 | 16 |
| 1 | 56.16 | 66.23 | 71.20 | 73.41 | | 1 | **26.94** | **27.82** | **29.34** | **29.74** |
| 5 | 58.13 | 70.88 | 75.92 | 79.33 | | 5 | 24.92 | 25.94 | 27.76 | 28.90 |
| 10 | **64.49** | **70.54** | **77.25** | **80.34** | | 10 | 23.06 | 25.25 | 27.18 | 27.69 |
| 25 | 56.52 | 64.65 | 68.95 | 74.52 | | 25 | 20.30 | 22.73 | 24.73 | 26.20 |
| 100 | 23.83 | 32.28 | 39.14 | 46.12 | | 100 | 17.84 | 18.82 | 20.51 | 23.57 |

Table 3: Effect of Channel Inhibition. *Left*: mAP@All (%) on MNIST, *Right*: mAP@1000 (%) on CIFAR-10. The number active channels per spatial location is denoted by $k_{\text{CI}}$. It can seen that channel inhibition (high sparsity) is critical for good performance. Total number of available channels for each kernel size was 500 and 400 for MNIST and CIFAR-10 respectively.

(see appendix for details), see Figure 4. There is an optimal level of activity for each dataset. For MNIST and CIFAR-10, $a$ was set to $0.05$ and $0.005$ respectively for all experiments. We visualize tSNE embeddings for different settings of activity levels in Figure 5. Interestingly, at lower sparsity levels, dissimilar images may become nearest neighbors though highly dissimilar images stay apart. This is reminiscent of an experimental finding (Lin et al., 2014) in Drosophila. Sparsification of Kenyon cells in Drosophila is controlled by feedback inhibition from the anterior paired lateral neuron. Disrupting this feedback inhibition leads to denser representations, resulting in fruit flies being able to discriminate between dissimilar odors but not similar odors.

**Convolutional BioHash** In the case of MNIST, we trained 500 convolutional filters (as described in Sec. 2.4) of kernel sizes $K = 3, 4$. In the case of CIFAR-10, we trained 400 convolutional filters of kernel sizes $K = 3, 4$ and 10. The convolutional variant of `BioHash`, which we call `BioConvHash` shows further improvement over `BioHash` on MNIST as well as CIFAR-10, with even small hash lengths $k \in \{2, 4\}$ substantially outperforming other methods at larger hash lengths. Channel Inhibition is *critical* for performance of `BioConvHash` across both datasets, see Table 3. A high amount of sparsity is essential for good performance. As discussed previously discussed, convolutions in our network are atypical in yet another way, due to patch normalization. We show in Sec 6, that patch normalization results in robustness of `BioConvHash` to "shadows", a robustness also characteristic of biological vision.

**Hashing using deep CNN features** State-of-the-art hashing methods generally adapt deep CNNs trained on ImageNet (Su et al., 2018; Jin et al., 2019; Chen et al., 2018; Lin et al., 2017). These approaches derive large performance benefits from the semantic information learned in pursuit of the classification goal on ImageNet (Deng et al., 2009). To make a fair comparison with our work, we trained `BioHash` on features extracted from fc7 layer of VGG16 (Simonyan & Zisserman, 2014), since previous work (Su et al., 2018; Lin et al., 2015; Chen et al., 2018) has often adapted this pre-trained network. `BioHash` demonstrates substantially improved performance over recent deep unsupervised hashing methods with mAP@1000 of 63.47 for $k = 16$; example retrievals are shown in Figure 3. Even at very small hash lengths of $k \in \{2, 4\}$, `BioHash` outperforms other methods at $k \in \{16, 32, 64\}$. For performance of other methods and performance at varying hash lengths see Table 4.

| Method | Hash Length ($k$) | | | | | |
| --- | --- | --- | --- | --- | --- | --- |
| | 2 | 4 | 8 | 16 | 32 | 64 |
| LSH* | 13.25 | 17.52 | 25.00 | 30.78 | 35.95 | 44.49 |
| PCAHash* | 21.89 | 31.03 | 36.23 | 37.91 | 36.19 | 35.06 |
| FlyHash* | 25.67 | 32.97 | 39.46 | 44.42 | 50.92 | 54.68 |
| SH* | 22.27 | 31.33 | 36.96 | 38.78 | 39.66 | 37.55 |
| ITQ* | 23.28 | 32.28 | 41.52 | 47.81 | 51.90 | 55.84 |
| DeepBit | - | - | - | 19.4 | 24.9 | 27.7 |
| USDH | - | - | - | 26.13 | 36.56 | 39.27 |
| SAH | - | - | - | 41.75 | 45.56 | 47.36 |
| GreedyHash | 10.56 | 23.94 | 34.32 | 44.8 | 47.2 | 50.1 |
| NaiveBioHash* | 18.24 | 26.60 | 31.72 | 35.40 | 40.88 | 44.12 |
| BioHash* | **57.33** | **59.66** | **61.87** | **63.47** | **64.61** | - |

Table 4: mAP@1000 (%) on CIFAR-10CNN. Best results (second best) for each hash length are in **bold** (underlined). `BioHash` demonstrates the best retrieval performance, substantially outperforming other methods including deep hashing methods `GreedyHash`, `SAH`, `DeepBit` and `USDH`, especially at small $k$. Performance for `DeepBit`,`SAH` and `USDH` is unavailable for some $k$, since it is not reported in the literature. * denotes the corresponding hashing method using representations from VGG16 fc7.

It is worth remembering that while exact Hamming distance computation is $O(k)$ for all the methods under consideration, unlike classical hashing methods, `BioHash` (and also `FlyHash`) incurs a storage cost of $k \log_2 m$ instead of $k$ per database entry. In the case of MNIST (CIFAR-10), `BioHash` at $k = 2$ corresponds to $m = 40$ ($m = 400$) entailing a storage cost of 12 (18) bits respectively. Even in scenarios where storage is a limiting factor, `BioHash` at $k = 2$ compares favorably to other methods at $k = 16$, yet Hamming distance computation remains cheaper for `BioHash`.

## 4    CONCLUSIONS, DISCUSSION, AND FUTURE WORK

Inspired by the recurring motif of sparse expansive representations in neural circuits, we introduced a new hashing algorithm, `BioHash`. In contrast with previous work (Dasgupta et al., 2017; Li et al., 2018), `BioHash` is *both* a data-driven algorithm and has a reasonable degree of biological plausibility. `BioHash` demonstrates strong empirical results outperforming recent unsupervised deep hashing methods. The biological plausibility of our work provides support toward proposal that LSH might be a general computational function (Valiant, 2014) of the neural circuits featuring sparse expansive representations. From the perspective of computer science `BioHash` is easy to train and scalable to large datasets.

Compressed sensing/sparse coding have also been suggested as computational roles of sparse expansive representations in biology (Ganguli & Sompolinsky, 2012). These ideas, however, require that the input be reconstructable from the sparse latent code. This is a much stronger assumption than LSH - downstream tasks might not require such detailed information about the inputs, e.g: novelty detection (Dasgupta et al., 2018).

In this work, we limited ourselves to linear scan using fast Hamming distance computation for image retrieval, like much of the relevant literature (Dasgupta et al., 2017; Su et al., 2018; Lin et al., 2015; Jin, 2018). Yet, there is potential for improvement. One line of future inquiry would be to speed up retrieval using multi-probe methods, perhaps via psuedo-hashes (Sharma & Navlakha, 2018). Another line of inquiry would be to adapt `BioHash` for Maximum Inner Product Search (Shrivastava & Li, 2014; Neyshabur & Srebro, 2015).

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

## SUPPLEMENTARY MATERIAL

We expand on the discussion of related work in section 5. We also include here some additional results. In section 6, we show that `BioConvHash` enjoys robustness to intensity variations. In section 7, we show that the strong empirical performance of `BioHash` is not specific to the choice of VGG16. Finally, we include technical details about implementation and architecture in section 8.

## 5 APPENDIX A: RELATED WORK

**Sparse High-Dimensional Representations in Neuroscience.** Previous work has explored the nature of sparse high-dimensional representations through the lens of sparse coding and compressed sensing (Ganguli & Sompolinsky, 2012). Additionally, Sompolinsky (2014) has examined the computational role of sparse expansive representations in the context of categorization of stimuli as appetitive or aversive. They studied the case of random projections as well as learned/"structured" projections. However, structured synapses were formed by a Hebbian-like association between each cluster center and a corresponding fixed, randomly selected pattern from the cortical layer; knowledge of cluster centers provides a strong form a "supervision"/additional information, while BioHash does not assume access to such information. To the best of our knowledge no previous work has systematically examined the proposal that LSH maybe a computational principle in the brain in the context of structured synapses learned in a biologically plausible manner.

**Classical LSH.** A classic LSH algorithm for angular similarity is `SimHash` (Charikar, 2002), which produces hash codes by $h(x) = \text{sign}(W^{\mathsf{T}}x)$, where entries of $W \in \mathbb{R}^{m \times d}$ are i.i.d from a standard normal distribution and sign() is element-wise. While `LSH` is a property and consequently is sometimes used to refer to hashing methods in general, when the context is clear we refer to `SimHash` as `LSH` following previous literature.

**Fruit Fly inspired LSH.** The fruit fly's olfactory circuit has inspired research into new families (Dasgupta et al., 2018; Sharma & Navlakha, 2018; Li et al., 2018) of Locality Sensitive Hashing (LSH) algorithms. Of these, `FlyHash` (Dasgupta et al., 2017) and `DenseFly` (Sharma & Navlakha, 2018) are based on random projections and cannot *learn* from data. Sparse Optimal Lifting (`SOLHash`) (Li et al., 2018) is based on learned projections and results in improvements in hashing performance. `SOLHash` attempts to learn a sparse binary representation $Y \in \mathbb{R}^{n \times m}$, by optimizing

$$\underset{\substack{Y \in [-1,1]^{n \times m} \\ Y\mathbf{e}_d = (-m+2k)\mathbf{e}_m}}{\arg\min} \quad ||XX^{\mathsf{T}} - YY^{\mathsf{T}}||_F^2 + \gamma||Y||_p, \tag{13}$$

$\mathbf{e}_m$ is an all 1's vector of size $m$. Note the relaxation from a binary $Y \in \{-1, 1\}^{n \times m}$ to continuous $Y \in [-1, 1]^{n \times m}$. After obtaining a $Y$, queries are hashed by learning a linear map from $X$ to $Y$ by minimizing

$$\underset{\substack{W \in [-1,1]^{d \times m} \\ W\mathbf{e}_d = (-m+2c)\mathbf{e}_m}}{\arg\min} \quad ||Y - XW||_F^2 + \beta||W||_p, \tag{14}$$

Here, $c$ is the # of synapses with weight 1; the rest are $-1$. To optimize this objective, Li et al. (2018) resorts to Franke-Wolfe optimization, wherein every learning update involves solving a constrained linear program involving all of the training data, which is biologically unrealistic. In contrast, `BioHash` is neurobiologically plausible involving only Hebbian/Anti-Hebbian updates and inhibition.

From a computer science perspective, the scalability of `SOLHash` is highly limited; not only does every update step invoke a constrained linear program but the program involves pairwise similarity matrices, which can become intractably large for datasets of even modest size. This issue is further exacerbated by the fact that $m \gg d$ and $YY^{\mathsf{T}}$ is recomputed at every step (Li et al., 2018). Indeed, though Li et al. (2018) uses the SIFT1M dataset, the discussed limitations limit training to only 5% of the training data. Nevertheless, we make a comparison to `SOLHash` in Table 5 and see that `BioHash` results in substantially improved performance.

**Deep LSH.** A number of state-of-the-art approaches (Su et al., 2018; Jin, 2018; Do et al., 2017; Lin et al., 2015) to unsupervised hashing for image retrieval are perhaps unsurprisingly, based on deep CNNs trained on ImageNet Deng et al. (2009); A common approach (Su et al., 2018) is to adopt a pretrained DCNN as a backbone, replace the last layer with a custom hash layer and objective

| Method | Hash Length ($k$) | | | | | |
|--------|------|------|------|------|------|------|
|        | 2    | 4    | 8    | 16   | 32   | 64   |
| BioHash | **39.57** | **54.40** | **65.53** | **73.07** | **77.70** | **80.75** |
| SOLHash | 11.59 | 20.03 | 30.44 | 41.50 | 51.30 | - |

Table 5: mAP@100 (%) on MNIST, using Euclidean distance in pixel space as the ground truth, following protocol in Li et al. (2018). BioHash demonstrates the best retrieval performance, substantially outperforming SOLHash.

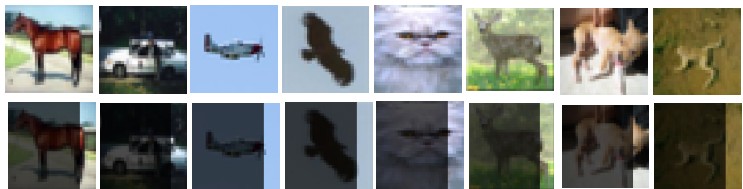

Figure 6: Examples of images with and without a "shadow". We modified the intensities in the query set of CIFAR-10 by multiplying 80% of each image by a factor of 0.3; such images largely remain discriminable to human perception.

function and to train the network by backpropogation. Some other approaches (Yang et al., 2018), use DCNNs as feature extractors or to compute a measure of similarity in it's feature space, which is then used as a training signal. While deep hashing methods are not the purpose of our work, we include them here for completeness.

## 6 APPENDIX B: ROBUSTNESS OF BIOCONVHASH TO VARIATIONS IN INTENSITY

Patch normalization is reminiscent of the canonical neural computation of divisive normalization (Carandini & Heeger, 2011) and performs local intensity normalization. This makes BioConvHash robust to variations in light intensity. To test this idea, we modified the intensities in the query set of CIFAR-10 by multiplying 80% of each image by a factor of 0.3; such images largely remain discriminable to human perception, see Figure 6. We evaluated the retrieval performance of this query set with "shadows", while the database (and synapses) remain unmodified. We find that BioConvHash performs best at small hash lengths, while the performance of other methods except GreedyHash is almost at chance. These results suggest that it maybe beneficial to incorporate divisive normalization into DCNNs architectures to increase robustness to intensity variations.

| Method | Hash Length ($k$) | | | | | |
|--------|------|------|------|------|------|------|
|        | 2    | 4    | 8    | 16   | 32   | 64   |
| LSH | 10.62 | 11.82 | 11.71 | 11.25 | 11.32 | 11.90 |
| PCAHash | 10.61 | 10.60 | 10.88 | 11.33 | 11.79 | 11.83 |
| FlyHash | 11.44 | 11.09 | 11.86 | 11.89 | 11.45 | 11.44 |
| SH | 10.64 | 10.45 | 10.45 | 11.70 | 11.26 | 11.30 |
| ITQ | 10.54 | 10.68 | 11.65 | 11.00 | 10.95 | 10.94 |
| BioHash | 11.05 | 11.50 | 11.57 | 11.33 | 11.59 | - |
| BioConvHash | **26.84** | **27.60** | **29.31** | 29.57 | 29.95 | - |
| GreedyHash | 10.56 | 21.47 | 25.21 | **30.74** | **30.16** | **37.63** |

Table 6: Robustness to shadows. mAP@1000 (%) on CIFAR-10 (higher is better), when query set has "shadows". Performance of other hashing methods drops substantially, while the performance of BioConvHash remains largely unchanged due to patch normalization. For small $k$, BioConvHash substantially outperforms all the other methods, while still being competitive at higher hash lengths. Best results (second best) for each hash length are in **bold** (underlined).

## 7 APPENDIX C: EVALUATION USING VGG16BN AND ALEXNET

The strong empirical performance of `BioHash` using features extracted from VGG16 fc7 is not specific to choice of VGG16. To demonstrate this, we evaluated the performance of `BioHash` using VGG16 with batch normalization (BN) (Ioffe & Szegedy, 2015) as well as AlexNet (Krizhevsky et al., 2012). Consistent with the evaluation using VGG16 reported in the main paper, `BioHash` consistently demonstrates the best retrieval performance, especially at small $k$.

| Method | Hash Length ($k$) | | | | | |
|---|---|---|---|---|---|---|
| | 2 | 4 | 8 | 16 | 32 | 64 |
| LSH | 13.16 | 15.86 | 20.85 | 27.59 | 38.26 | 47.97 |
| PCAHash | 21.72 | 34.05 | 38.64 | 40.81 | 38.75 | 36.87 |
| FlyHash | 27.07 | 34.68 | 39.94 | 46.17 | 52.65 | 57.26 |
| SH | 21.76 | 34.19 | 38.85 | 41.80 | 42.44 | 39.69 |
| ITQ | 23.02 | 34.04 | 44.57 | 51.23 | 55.51 | 58.74 |
| BioHash | **60.56** | **62.76** | **65.08** | **66.75** | **67.53** | - |

Table 7: mAP@1000 (%) on CIFAR-10CNN, VGG16BN. Best results (second best) for each hash length are in **bold** (underlined). `BioHash` demonstrates the best retrieval performance, especially at small $k$.

| Method | Hash Length ($k$) | | | | | |
|---|---|---|---|---|---|---|
| | 2 | 4 | 8 | 16 | 32 | 64 |
| LSH | 13.25 | 12.94 | 18.06 | 23.28 | 25.79 | 32.99 |
| PCAHash | 17.19 | 22.89 | 27.76 | 29.21 | 28.22 | 26.73 |
| FlyHash | 18.52 | 23.48 | 27.70 | 30.58 | 35.54 | 38.41 |
| SH | 16.66 | 22.28 | 27.72 | 28.60 | 29.27 | 27.50 |
| ITQ | 17.56 | 23.94 | 31.30 | 36.25 | 39.34 | 42.56 |
| BioHash | **44.17** | **45.98** | **47.66** | **49.32** | **50.13** | - |

Table 8: mAP@1000 (%) on CIFAR-10CNN, AlexNet. Best results (second best) for each hash length are in **bold** (underlined). `BioHash` demonstrates the best retrieval performance, especially at small $k$.

## 8 APPENDIX D: IMPLEMENTATION DETAILS

- `BioHash`: The training /retrieval database was centered. Queries were also centered using mean computed on the training set. Weights were initialized by sampling from the standard normal distribution. For simplicity we used $p = 2, \Delta = 0$. We set initial learning rate $\epsilon_0 = 2 \times 10^{-2}$, which was decayed as $\epsilon_t = \epsilon_0(1 - \frac{t}{T_{\max}})$, where $t$ is epoch number and $T_{\max}$ is maximum number of epochs. We used $T_{\max} = 100$ and a mini-batch size of 100. The criterion for convergence was average norm of synapses was $< 1.06$. Convergence usually took $< 20$ epochs.

  In order to set the activity level, we performed cross-validation. In the case of MNIST, we separated 1k random samples (100 from each class) from the training set, to create a training set of size 68k and validation set of 1k images. Activity level with highest mAP@All on the validation set was determined to be 5%, see Figure 4. We then retrained `BioHash` on the whole training data of size 69k and reported the performance on the query set. Similarly for CIFAR-10, we separated 1k samples (100 images per class) to create a training set of size 49k and validation set of 1k. We set the activity level to be 0.5%, see Figure 4. We then retrained `BioHash` on the whole training data of size 50k and reported the performance on the query set.

- `BioConvHash` A convolutional filter of kernel size $K$ is learned by dividing the training set into patches of sizes $K \times K$ and applying the learning dynamics 1. In the case of MNIST, we trained 500 filters of kernel sizes $K = 3, 4$. In the case of CIFAR-10, we used $K = 3, 4, 10$. For both datasets, we used a stride of 1 in the convolutional layers. We set $k_{\text{CI}} = 10$ for MNIST and $k_{\text{CI}} = 1$ for CIFAR-10 , by cross-validation in a procedure similar

to how sparsity was set. The effect of channel inhibition is shown in Table 3. $k_{CI} = 1$ means that only the largest activation across channels per spatial location was kept, while the rest are set to 0. This was followed by 2d max-pooling with a stride of 2 and kernel size of 7. This was followed by a fully connected layer (the "hash" layer).

- `FlyHash` Following Dasgupta et al. (2017), we set $m = 10d$ for all hash lengths $k$ and each neuron in the hashing layer ("Kenyon" cell) sampled from 0.1 dimensions of input data (Projection neurons). Following Gong & Lazebnik (2011), `ITQ` employed 50 iterations.

- To extract representations from VGG16 fc7, CIFAR-10 images were resized to $224 \times 224$ and normalized using default values: $[0.485, 0.456, 0.406]$, $[0.229, 0.224, 0.225]$. To make a fair comparison we used the pre-trained VGG16 model (without BN), since this model is frequently employed by deep hashing methods. We also evaluated the performance using VGG16 with BN and also using AlexNet (Krizhevsky et al., 2012), see Tables 7, 8.

- `GreedyHash` replaces the softmax layer of VGG16 with a hash layer and is trained end-to-end via backpropogation using a custom objective function, see Su et al. (2018) for more details. We use the code [3] provided by the authors to measure performance at $k = 2, 4, 8$, since these numbers were not reported in Su et al. (2018). We used the default parameters: mini-batch size of 32, learning rate of $1 \times 10^{-4}$ and trained for 60 epochs.

## 9 APPENDIX E: TRAINING TIME

Here we report the training times for the best performing (having the highest corresponding mAP@R) variant of our algorithm: `BioHash`, `BioConvHash`, or `BioHash` on top of VGG16 representations. For the case of MNIST, the best performing variant is `BioConvHash`, and for CIFAR-10 it is `BioHash` on top of VGG16 representations. We also report the training time of the next best method for each dataset. This is `GreedyHash` in the case of CIFAR-10, and `BioHash` in the case of MNIST. In the case of MNIST, the best method that is not a variant of `BioHash` is `UH-BNN`. Training time for `UH-BNN` is unavailable, since it is not reported in literature. All experiments were run on a single V100 GPU to make a fair comparison.

| Method | Hash Length ($k$) | | | | |
|---|---|---|---|---|---|
| | 2 | 4 | 8 | 16 | 32 |
| `BioHash` | ~1.7 s | ~1.7s | ~1.7 s | ~3.4 s | ~ 5 s |
| `BioConvHash` | ~3.5 m | ~3.5 m | ~3.5 m | ~5 m | ~5 m |

Table 9: Training time for the best variant of `BioHash`, and the next best method for MNIST.

| CIFAR-10 | Hash Length ($k$) | | | | |
|---|---|---|---|---|---|
| | 2 | 4 | 8 | 16 | 32 |
| `BioHash` | ~ 4.2 s | ~7.6 s | ~11.5 s | ~22 s | ~35 s |
| `GreedyHash` | ~1.2 hrs | ~1.2 hrs | ~1.3 hrs | ~1.4 hrs | ~1.45 hrs |

Table 10: Training time for the best variant of `BioHash` and the next best method for CIFAR-10. Both models are based on VGG16.

---

[3] https://github.com/ssppp/GreedyHash

