# OpenReview forum: "Bio-Inspired Hashing for Unsupervised Similarity Search"
_ICLR.cc/2020/Conference — Reject_

### Official Review · AnonReviewer3 · 2019-10-23
**Official Blind Review #3**

**Rating:** 6

**Review:**

This paper introduces a variant of FlyHash for similarity search in vector space. The basic idea is motivated by the intuition: the original FlyHash method is data-independent, so can we improve FlyHash's locality sensitivity by learning from data. It does so by learning the weights of the projection layer and uses winner-take-all sparsification to generate sparse binary hash code. This leads to the bio-inspired hashing algorithm (BioHash).  The paper argues that by taking into account the density of data, the learned projection helps to further push similar entities to have hash codes that point in similar direction while repelling dissimilar objects in opposite directions. Experiment results are reported on MNIST and CIFAR10, and the proposed approach demonstrates better retrieval precision results compared to several other hashing based methods.

Strengths:
+ A good example of intersection work that has interesting indications to both the biological and computational sides.
+The unsupervised learning method for sparse representation expansion seems to be novel.

Weaknesses:
- Interesting exploration but still seems to have a large gap to a real-world solution (see comments below).
- Sometimes the writing is a bit hard to follow, presumably because it introduces the work using concepts from both fields.

Overall, I like the idea of this paper. In particular, I find the reverse indication that learned synapses must be neurobiologically plausible really interesting. It is also a sensible approach to learn from data to improve the locality sensitivity of FlyHash than doing just random projection. However, I feel there is still a gap between the current work and the real working system.

First, the paper only studies the accuracy impact of BioHash but completely ignores the evaluation of search time and memory, which are crucial dimensions for evaluating similarity search algorithms. Without those constraints, similarity search can just be done with brute force searches without any representation transformation.  By using accuracy as a single metric, it is tricky to get a sense of how good the proposed solution is.  It might outperform FlyHash and other hashing based approach, but a caveat is that hashing itself has not been identified as the most effective way for performing similarity search.

Second, the evaluation is mostly done on toy datasets in terms of scale. The state-of-the-art similarity search is often evaluated on millions and sometimes billion-scale datasets, such as BIGANN and DEEP1B. Given that the proposed approach is only evaluated on MNIST and CIFAR10, it is unclear how scalable the proposed solution is on larger datasets. Empirical studies have shown that hashing based approaches often incur a large accuracy loss for dense continuous vectors. On million-scale datasets, solutions such as similarity graph-based approaches (e.g., HNSW) outperform hashing and quantization based approaches by a large margin. It could be that through data-driven sparse expansion, bio-inspired hashing could help overcome the limitation of existing hashing based approaches, but without comparison, we do not know the answer.

Third, I thought the paper neglects many practical aspects of doing a similarity search. For example, no indexing is applied to the generated binary hash code and the search is done through linear scanning. There is also no discussion on training time and how to handle incremental updates.

Another minor issue of this paper is that the citation format does not seem to comply with ICLR format. But I believe these issues could be easily fixed.

That being said, while the method does not seem to be readily applicable to real-world scenarios and only manages to demonstrate its effectiveness on small datasets compared to other hashing based approaches, I think this is a promising direction to look into, and I imagine it could serve as a starting point for other researchers to develop more extensions on top of it.


**Experience Assessment:**

I have published one or two papers in this area.

**Review Assessment: Checking Correctness Of Derivations And Theory:**

I assessed the sensibility of the derivations and theory.

**Review Assessment: Checking Correctness Of Experiments:**

I assessed the sensibility of the experiments.

**Review Assessment: Thoroughness In Paper Reading:**

I read the paper at least twice and used my best judgement in assessing the paper.

---

### Official Review · AnonReviewer1 · 2019-10-26
**Official Blind Review #1**

**Rating:** 6

**Review:**

This paper studies a new model of locally sensitive hashing (LSH) that is inspired by the fruit fly Drosophila's olfactory circuit. Instead of mapping each input to a low-dimensional space, such LSH methods (FlyHash) map given $d$-dimensional inputs to an $m$-dimensional space such that $m \gg d$. However, these mappings enforce sparsity on the hash maps such that only $k$ out of $m$ coordinates of the output of the hash map are non-zero.

The existing methods to realize these FlyHash maps rely on random projections and don't take the original dataset being mapped by the hash function into account. This paper proposes a data-driven approach to learn such sparsity inducing hash maps which are referred to as BioHash. The proposed approach is biologically feasible, which raises the interesting question if various biological organisms employ such a mechanism to realize some kind of LSH.

Using MNIST and CIFAR-10 datasets, the paper shows that BioHash achieves superior performance as a similarity search mechanism as compared to many existing LSH methods for small values of $k$. The paper also evaluates a convolutional variant of BioHash, namely BioConvHash, which leads to even better similarity search scores. This makes BioHash and BioConvHash an appealing candidate to realize fast and scalable similarity searches on large scale datasets.

Overall, the paper makes some interesting contributions with experimental results clearly showing the utility of the proposed method. That said, there is some room for improvement in the presentation of the key ideas.

- What is $Rank(\cdot)$ in Eq. 1 and Eq. 3?

- $W_{\mu}$ is not formally defined ($W_{\mu} = (W_{\mu 1},\ldots, W_{\mu d})$?)

- 'figure', 'table', 'sec.' --> 'Figure', 'Table', 'Sec.'.

- Notation mAP@x is not formally defined in Section 3.1 and 3.2. Also, is there notation overloading. How are $k$ defining the sparsity in the map and $k$ in Section 3.1 related?

- In Section 5 (appendix), the authors point out that data-driven Fly inspired LSH exists in the literature (SolHash by Li et al.) Even though SolHash is not a biological feasible approach, it somewhat contradicts one of the underlying claims in the main text that this paper is the first paper to propose data-driven Fly inspired hash maps. Shouldn't the authors mention SolHash in the main text itself and introduce their contributions in the proper context?

------------------------------------------
Post rebuttal phase
------------------------------------------

Thank you for addressing my presentation related questions. With improved discussion of prior work, the paper places its contributions in the right context. Moreover, the empirical results in the paper demonstrate the utility of the proposed method on (small scale) real world data. This should inspire a thorough investigation on large scale data sets in the future. Overall, I think that this paper makes interesting and novel contributions. Accordingly, I have updated my score to Weak Accept.

**Experience Assessment:**

I have published one or two papers in this area.

**Review Assessment: Checking Correctness Of Derivations And Theory:**

I assessed the sensibility of the derivations and theory.

**Review Assessment: Checking Correctness Of Experiments:**

I carefully checked the experiments.

**Review Assessment: Thoroughness In Paper Reading:**

I read the paper at least twice and used my best judgement in assessing the paper.

---

### Official Review · AnonReviewer2 · 2019-10-28
**Official Blind Review #2**

**Rating:** 3

**Review:**

This paper introduces a bio-inspired locally sensitive hashing method called BioHash, inspired by FlyHash. Unlike FlyHash and many other hashing algorithms, the BioHash algorithm is adaptive and uses synaptic plasticity to update the mapping from input to encoding.
The experimental section uses common datasets (CIFAR-10 & MNIST) to compare the performance of BioHash to a number of common locally sensitive hashing methods and BioHash is shown to significantly outperform existing method in similarity search tasks.

Caveat: I'm not an expert in this field and did my best to understand the details. I'm not familiar with the the state-of-the-art of hashing methods.

Questions:
- Novelty of the algorithm? As the authors point out ("We adopt a biologically plausible unsupervised algorithm for representation learning from (17)."), the algorithm comes from another recent paper (17, "Unsupervised learning by competing hidden units", PNAS 2019). What is the difference between the PNAS 2019 paper and this submission? It seems like the algorithm from [17] was directly applied to similarity search, without additional contributions.
- Overall, I'm having a hard time understanding the equations in section 2.1. This is the main reason why I read [17]. In equation 1, you use square brackets for the input of a function g[...], but in equation 2, you use parenthesis (expected). I also had to refer to [17] to figure out what "Rank" meant in this context (hint: it's not the rank of a matrix).
- Section 2.2 "Intuition behind the learning algorithm" doesn't clear things up. I think a visual representation of the learning method would better serve this purpose.

**Experience Assessment:**

I do not know much about this area.

**Review Assessment: Checking Correctness Of Derivations And Theory:**

I did not assess the derivations or theory.

**Review Assessment: Checking Correctness Of Experiments:**

I did not assess the experiments.

**Review Assessment: Thoroughness In Paper Reading:**

I read the paper at least twice and used my best judgement in assessing the paper.

---

### Decision · Program_Chairs · 2019-12-19

**Decision:**

Reject

**Comment:**

This paper introduces a biologically inspired locally sensitive hashing method, a variant of FlyHash. While the paper contains interesting ideas and its presentation has been substantially improved from its original form during the discussion period, the paper still does not meet the quality bar of ICLR due to its limitations in terms of experiments and applicability to real-world scenarios.

---

> ### Author Response · Authors · 2019-12-22
> **Response to the meta-review - clarification of  some misunderstandings.**
>
> We thank the AC and reviewers for their time and hard work.
>
> Having said that, we are disappointed and confused by the meta-review: “the paper still does not meet the quality bar of ICLR due to its limitations in terms of experiments and applicability to real-world scenarios.” Since no detail is provided, we wish to clarify apparent misunderstandings - at least for future readers.
>
> First, some context. In the unsupervised similarity search literature, broadly there are two settings - A) descriptors are provided, ground truth is based on Euclidean distance in descriptor space (e.g: BIGANN and DEEP1B) and B) descriptors are *learned* together with a hash map and ground truth is based on *semantic labels*. As explained in the rebuttal, our work belongs to setting B, similarly to Lin et al, 2015; Lu et al, 2017; Do et al, 2016; Do et al, 2017; Jin et al 2019; Su et al, 2018.
>
> - ”limitations in terms of experiments”: Since we include systematic ablations demonstrating the benefit of each component we introduce, we assume this refers to (lack of) indexing. As explained in the rebuttal, indexing is very atypical in setting B, though entirely common in setting A. Given this, it is not at all clear what are relevant comparisons the AC thinks are necessary to include.
>
>
> - ”applicability to real-world scenarios.” For context, FlyHash (Dasgupta et al, Science, 2017) and SOLHash (Li et al, NeurIPS 2018) both also inspired by the fruit fly’s olfactory circuit use datasets that are between *5x-7x* smaller than the ones considered in our work. Even more recently, Liu et al, NeurIPS 2019 (an oral presentation) uses datasets that range from ~5k-27k images, between 2x-10x smaller. (Note also, none of these use indexing). None of these (or any of the literature cited above) are real-world solutions, nor are they directly meant to be - nevertheless we find them interesting and useful contributions to the literature.  We find this feedback doubly confusing given that no citations of real-world scenarios in this literature (setting B) were suggested. Note that while some of this literature (e.g: Su et al, 2018) evaluates performance on a subset of ImageNet, this is generally a *supervised* hashing setting, while we consider the *unsupervised* setting.
>
> - It is also unfortunate that our work was treated simply as yet another hashing method, ignoring the biological plausibility of our method and contributions to neuroscience.